# Unpaid caregiving and mental health during the COVID-19 pandemic—A systematic review of the quantitative literature

**Jennifer Ervin**[ORCID]**\***, **Ludmila Fleitas Alfonzo, Yamna Taouk, Humaira Maheen, Tania King**

Centre for Health Policy, Melbourne School of Population and Global Health, The University of Melbourne, Parkville, Victoria, Australia

\* jennifer.ervin@unimelb.edu.au

## Abstract

The COVID-19 pandemic imposed additional and specific challenges on the lives and well-being of informal unpaid carers. Addressing an important gap in the existing literature, this systematic review (prospectively registered with PROSPERO CRD42022376012) synthesises and evaluates the quantitative evidence examining the association between unpaid caregiving and mental health (compared to non-caring), during the pandemic. Five databases were searched (Medline, PsycInfo, EMBASE, Scopus, Web of Science) from Jan 1, 2020, to March 1, 2023. Population-based, peer-reviewed quantitative studies using any observational design were included, with screening, data extraction and quality assessment (amended NOS) independently conducted by two reviewers. Of the 3,073 records screened, 20 eligible studies (113,151 participants) were included. Overall quality of evidence was moderate. Narrative synthesis was complemented by Effect-direction and Albatross plots (given significant between-study heterogeneity precluded meta-analysis). Results indicate that the mental health of informal carers, already poorer pre-COVID compared to non-caregivers, was disproportionally impacted as a result of the pandemic and its associated public health containment measures. This review highlights the vulnerability of this group and should motivate political will and commensurate policies to ensure unpaid caregivers are better supported now, in the medium term, and crucially if, and when, another global public health emergency emerges.

## 1. Introduction

Unpaid informal care contributes to an estimated 70–95% of the world's care needs, and significantly impacts the lives of those who provide it, imposing both positive and negative effects [1–3]. Moreover, unpaid care is deeply gendered, with women accounting for an estimated 80% of informal carers globally [4]. The COVID-19 pandemic highlighted not only the importance and scale of unpaid care provision but catapulted it to being an essential frontline component of the pandemic response across the globe, with demands increasing exponentially as public services were restricted [5, 6]. Ultimately the pandemic resulted in greater unpaid

**Data Availability Statement:** All relevant data are within the paper and its Supporting Information files.

**Funding:** This work was supported by a Melbourne School of Population and Global Health Targeted Research Support Grant and an Australian Research Council (ARC) Linkage Project (LP180100035), both awarded to TK. TK is supported by an ARC Discovery Early Career Award (DE200100607). The funders did not have any involvement in the design or conduct of any part of the study.

**Competing interests:** The authors have declared that no competing interests exist.

caring responsibilities and increased caregiving intensity for carers worldwide, the effects of which are only just starting to be understood. This review addresses a gap in the existing literature by synthesising and evaluating the quantitative evidence examining the association between informal unpaid caregiving and mental health (compared to non-caring), during the COVID-19 pandemic.

Mental health (MH) disorders are a considerable public health concern worldwide. Even prior to the pandemic, MH disorders (depression and anxiety in particular) were persistently among the top causes of the global disease burden [7]. In creating additional and unprecedented hazards to mental health, the pandemic triggered relatively higher rates of stress, anxiety, depression, and psychological distress across the globe [8, 9]. All major emergencies and disasters generate high levels of fear, uncertainty, and stress, and stress is an important pathway through which our social and physical environments affect our MH [10, 11]. Extensive literature also acknowledges the stressful nature of caregiving [12], and the caregiver stress process through which caregiving is theorised to affect MH [13]. This is supported by several (pre-COVID) reviews reporting a negative association between unpaid caregiving and MH for carers of all ages [14–20]. Compared to non-carers, caregivers were likely disproportionately affected over the course of the pandemic due to the additional (and interwoven) emotional, health-based, and financial stressors they experienced [21]. Yet, whilst much initial attention and focus were directed toward the paid care workforce, unpaid caregivers remained largely hidden in comparison [22]. Fortunately, increasing attention and recognition of the experiences of informal caregivers has emerged, and the emerging literature base reflects this.

The pandemic was a globally unifying experience and, whilst different countries experienced differing length and severity of public health measures, strict lockdowns were commonplace, as were strict quarantine measures and reduced service access. Such measures resulted in additional stressors for caregivers. Not only did reduced access to support services (e.g., home health assistance, allied health care, respite care) increase caring intensity/responsibilities, lockdowns severely disrupted protective social support networks and connections [21, 23]. Moreover, the danger of a life-threatening virus only intensified caregiver worries regarding care recipient health [24], often prompting further isolation due to fear of contracting and spreading COVID-19 to their loved one [21, 23]. Additionally, many caregivers also had to negotiate new or intensified economic stressors or job insecurities because of the pandemic [21]. In contrast, some positive effects have been postulated, including deeper connections with care recipients [25], spending more time with loved ones [26], and having a greater sense of purpose [27]. Therefore, the possible MH effects for caregivers (compared to non-carers) in the context of the pandemic are potentially mixed.

Despite our proximity to the pandemic, several reviews have already been published in this space. A rapid systematic review examined the MH impact of the pandemic on caregivers of adult family members [28]. From the eight included studies (screening ceasing in September 2021), the review reported two main themes: family caregivers COVID-19 related stress, and (mal) adaptive strategies to the "new" normality [28]. Meanwhile, a 2021 systematic review examined the MH of caregivers of dementia patients specifically, reporting worsening mental health and increased burden because of the COVID-19 pandemic and its associated public health measures [29]. Lastly, another systematic review examined the psychological impact of the pandemic (lockdown and quarantine measures) on caregivers of children and adolescents (<18yrs), reporting in pooled estimates that 52.3% and 27.4% of caregivers developed anxiety and depression, respectively, while being in isolation with children [30].

Whilst the above evidence provides an important starting point to understanding how the global pandemic has impacted the MH of unpaid caregivers, no reviews to date have examined the effect of caregiving *compared* to non-caregiving within this context. This is important

because the pandemic exerted a significant impact on the MH of the population. Population studies have demonstrated that MH among adults [31] and children [32] declined, with effects more pronounced in some subgroups. Without a comparator, it is not possible to distinguish the effects of the pandemic from the effects of caring. In addition, compared to extant reviews, our review criteria encompass the breadth of what constitutes unpaid caregiving, rather than being restricted to care for older adults, or care for children, or for specific care recipients based on disease or condition. Our review addresses both gaps and extends to include all published work in this space up until March 2023. Lastly, understanding the impact of the pandemic on carers is not merely an exercise in assuaging curiosity, but it is of significant practical importance. Not only are informal care demands predicted to continue to increase, but COVID-19 will not be humanity's last experience of a pandemic. As such, gleaning what we can from this pandemic is vital in informing responses to future pandemics.

The main aims of this review are:

1. To summarise the quantitative evidence examining the association between informal unpaid caregiving and MH (compared to non-caring) in the context of the COVID-19 pandemic.

2. To assess the quality of the existing evidence.

## 2. Methods

### 2.1. Search strategy

This review was prospectively registered in PROSPERO CRD42022376012 and followed the Preferred Reporting Items for Systematic Review and Meta-Analysis (PRISMA) [33] guidelines (see S1 Checklist for PRISMA checklist). A three-tiered search strategy was developed in Medline (OVID) and subsequently employed across all databases. A complete list of search terms for each tier (including MeSH terms) and strategies for each database is in S1 File. Literature searches were conducted on five electronic databases: Medline (OVID), PsycINFO (OVID), EMBASE (OVID), Scopus and Web of Science, and were restricted to peer-reviewed material published between January 1st, 2020, and March 1st, 2023 (screening ceased March 7th, 2023).

This review was restricted to quantitative population-based studies of any observational design (i.e., prospective cohort, case–control, retrospective or cross-sectional) examining associations between unpaid care and MH during the COVID-19 pandemic. Our exposure was classified as unpaid caregiving for individuals who were either temporarily or permanently ill or physically and/or mentally disabled, and the elderly and children (before they are legal adults) during the pandemic [34]. Caregiving in a volunteer capacity (i.e., to someone not kin/known) or as part of a paid professional vocation were ineligible. Unpaid caring status could be measured at any point during pandemic, with MH outcomes measured at or after exposure to caregiving. Studies were only eligible for inclusion if there was a clear non-caring comparator. Studies that examined the pandemic as the exposure (i.e., a carer's MH before the pandemic compared to their MH during the pandemic) were excluded unless there was an estimate that compared carers with non carers during the pandemic. No restrictions were placed on the age of included populations, such that young carers (under 18-25yrs), as well as senior adults in caregiving roles, were eligible for inclusion.

MH was the outcome of interest. To be eligible, studies had to use a validated measure of MH symptomology (e.g., Kessler, CES-D and GHQ-12 instruments) assessing common MH disorders such as depression, anxiety, and psychological distress. Physician MH diagnoses of

depression or anxiety were also eligible for inclusion. However, severe, or psychotic mental illnesses (such as schizophrenia), were excluded as these do not reflect the stress related pathways through which unpaid caregiving may affect MH outcomes. Any measure of effect or association was permissible. Where the same dataset was used in multiple studies, the most relevant and/or recent study (with our exposure/outcomes of interest) covering the longest period/largest number of waves was included. No restrictions on geographical setting, or country-level socioeconomic development were imposed, however studies where full-text English translation could not be sourced were excluded.

A web-based tool to conduct systematic reviews, Covidence [35], was employed to screen search results exported from the five electronic databases. Two reviewers (JE and LFA) independently screened all articles (title/abstract and full text) for inclusion. Reviewers were blinded to each other's decisions throughout the screening and latter quality assessment process. Disagreements were resolved through discussion, and where not resolved, a third reviewer (TK and/or YT and/or HM) was consulted.

## 2.2. Data analysis

**2.2.1. Data extraction.** A data extraction template, constructed in Covidence [35], was utilised to capture and summarise the characteristics of included studies, including title, author, year of publication, study location, study design, population, sample size, characterisation of exposure, characterisation of the outcome, confounder adjustment, analytical approach, and measures of effect. Data extraction was conducted by one reviewer (JE) and cross-checked by a second reviewer (LFA) in Covidence and, following consensus, study data was exported and recorded in an excel spreadsheet.

**2.2.2. Quality assessment.** The quality of included studies was assessed using a modified version of the Newcastle-Ottawa Scale (NOS) [36, 37]. Based on the precedent of Lacey et al. [19], we modified the NOS for cohort studies to factor in non-response rates, distinguish between longitudinal and cross-sectional cohort studies, and to assess statistical analysis. Described in detail in S2 File, in evaluating study quality (or risk of bias), stars were awarded where merited across ten criteria: four selection criteria, two confounding criteria, and four outcome criteria. Studies that scored 8 or above were deemed to be at low risk of bias (RoB), studies that scored 5–7 were judged at moderate/some RoB, whilst studies that scored 4 or below were considered at high RoB.

**2.2.3 Data synthesis.** There were considerable differences in how different studies interrogated and categorised unpaid caregiving, as well as heterogeneity in statistical methods of analysis and study designs. There were also significant differences in nuance between studies depending on the country context and the timing of the data collection relative to the pandemic, lockdowns, and other public orders. As such, data synthesis by way of a meta-analysis was not possible. Thus, as per Cochrane recommendations [38], findings were consequently synthesised and presented using alternative methods. These included a narrative synthesis, an effect direction plot [39], Fisher's meta-analysis of combining p values [38], and albatross plots [40] (see S3 File for further details).

# 3. Results

## 3.1. Study characteristics

A total of 5,816 studies were identified by the search, from which 2,743 duplicates were removed, resulting in a total of 3,073 records being screened by title and abstract. From these, 145 records were assessed for eligibility through full-text screening. As per Fig 1 (PRISMA flowchart), 125 of these were excluded due to not meeting the eligibility criteria, resulting in a

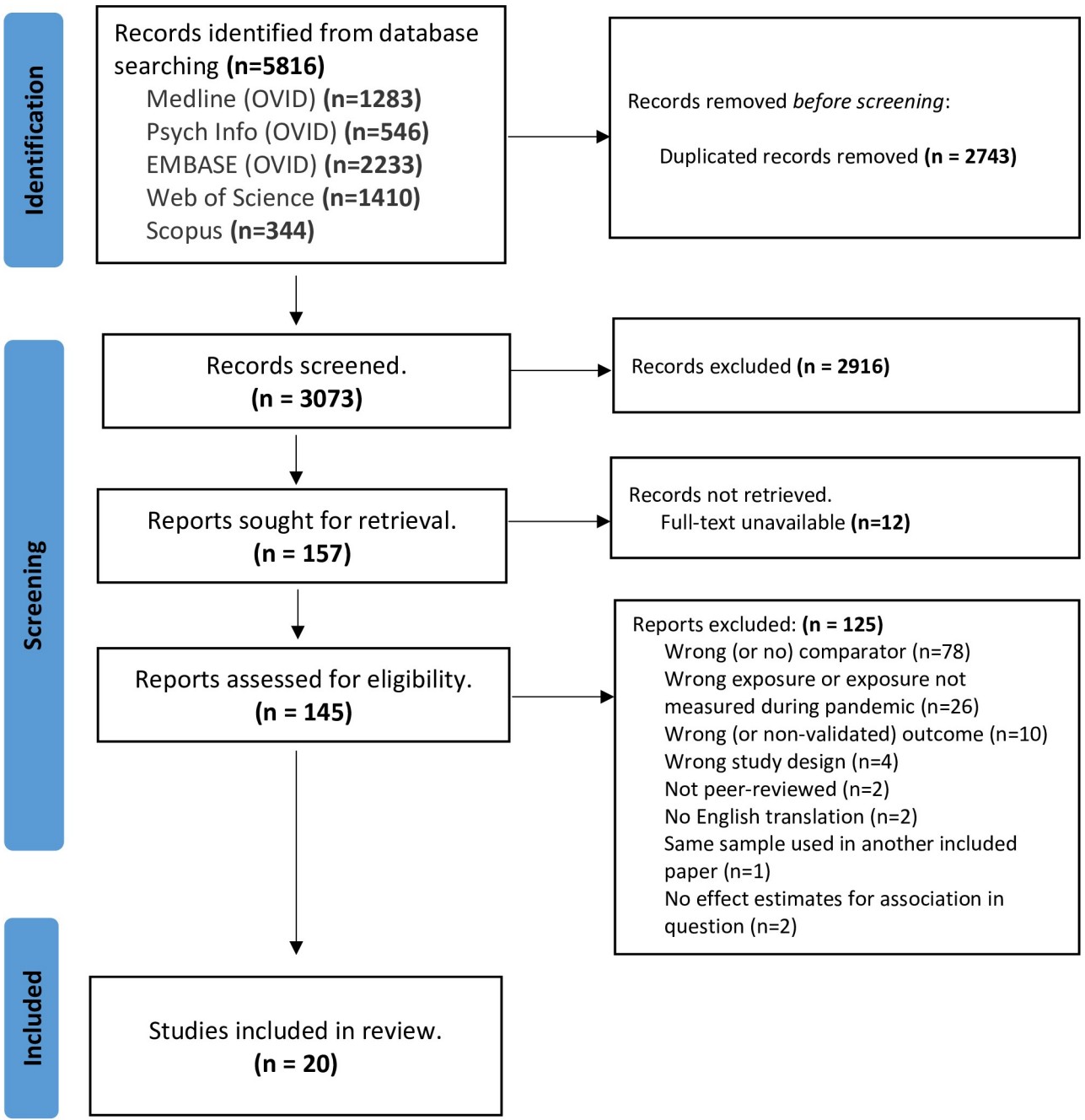

From: Page MJ, McKenzie JE, Bossuyt PM, Boutron I, Hoffmann TC, Mulrow CD, et al. The PRISMA 2020 statement: an updated guideline for reporting systematic reviews. BMJ 2021;372: n71. doi: 10.1136/bmj.n71

**Fig 1. Study selection diagram (PRISMA flowchart).**

total of 20 eligible studies (totalling 113,151 participants) for inclusion in the systematic review. S1 Table documents the studies excluded at full text review with reasons for exclusion, the most common reason being the absence of a non-caring comparator.

Of the included studies, six employed a longitudinal design [41–46], whilst fourteen were cross-sectional [47–60]. Geographically, there were five studies from the United Kingdom

[41, 42, 45, 46, 56], three from the US [44, 49, 55], three from Italy [48, 50, 60], two from Japan [57, 58], and one each from Australia [52], Austria [59], Canada [43], China [54], Hong Kong [53], Latin America [51], and New Zealand [47]. Whilst the majority of studies were sampled from the general population [41, 43–45, 49–52, 54–57], some studies were restricted to older adults [46, 47, 58], one was restricted to working age adults (20-64yrs) [59], another to young adults (18-29yrs) [60], one to undergraduate students [48], another to grandparents [42], and one study examined adult women only [53]. The studies varied in how they defined and analysed informal caregiving. The majority operationalised unpaid caregiving as a binary variable (care provision versus no care provision), whilst seven studies categorised caregiving in a variety of ways. Two studies categorised carers as new carers or continued/existing carers [41, 46], one study had a very broad definition of caregiving [46], whilst the other restricted their caring exposure to out-of-home caregiving [41]. One study divided carers into short-term and long-term (>12 months) caregivers [44], whilst another examined childcare and eldercare separately as well as together (sandwich care) [54]. Two studies inferred caregiving based on family/household composition. One study classified family caregivers if any child or adult in the household had a disability (presumed care) vs having a disability themselves, with the comparator group being no household members with disability (including self); the presumed non-caring comparator [51]. Another study (examining young adult carers) presumed caregiving based on who (if anyone) in their household had a serious physical or mental health condition (i.e., parent or other ill family member, rather than being explicitly self-reported [60]. The sixth study focussed on care among grandparents, examining changes in grandparental care because of the pandemic. Those doing mostly the same (or increased) grandparental care were classified as caregivers, and non-carers were those grandparents who were caring for grandchildren pre-pandemic and then had completely stopped caregiving since the pandemic began [42].

Of the thirteen studies that examined caregiving (vs non care) as binary exposure, nuance existed in how the exposure was measured/operationalised. Some studies defined caregiving as broadly providing unpaid care or assistance to anyone with a disability, chronic illness or who was elderly [43, 47, 59]. Other studies nominated family member caregiving specifically [49, 57, 58], whilst four studies only considered caregiving for someone residing in the same household [45, 48, 50, 53]. In addition to the grandparental care study [42], two studies specifically examined childcare, one classifying caregivers based solely on whether they were the primary carer for a child [55], and another examining two caregiving exposures separately [52], one being childcare and the other caregiving for dependant relatives.

Every included study utilised a validated measure of MH (in line with the review criteria), all of which were self-reported via survey-based instruments. Seven studies examined depression and anxiety as two separate MH outcomes [43, 47–49, 52, 53, 60], whilst twelve studies examined only depression/physiological distress [41, 42, 44–46, 50, 54–59], and one study examined only anxiety [51]. A descriptive summary and main findings of included studies (including effect estimates and confidence parameters where reported) can be found in Table 1.

### 3.2. Quality assessment

Table 1 reports the overall star rating for each of the included studies based on the NOS quality assessment. S2 File presents detailed NOS results, including the study-level star allocation for each domain of quality/bias considered. Overall, in evaluating the relationship between informal unpaid caregiving and MH in the context of the COVID-19 pandemic, the quality of evidence was found to be moderate. Five studies were deemed to be at low risk of bias (RoB) [41, 42, 44, 46, 47], twelve studies at moderate RoB [43, 45, 49–52, 54, 56–60], and three studies

**Table 1. Descriptive summary of included studies.**

| First author (year) | Country | Analytical sample | Study design (for our estimate of interest) and data source | Exposure/s and measurement | Mental health outcomes | Main findings (All estimates reported are from adjusted models where available) | Quality assessment (NOS) |
|---|---|---|---|---|---|---|---|
| Allen 2022 | New Zealand | Older adults 55–81yrs 3839 participants (718 caregivers (CG), 3121 non-carers (NC)) | Cross-sectional Data from older adults participating in the 2020 Health, Work and Retirement longitudinal survey | *Caregiving* Participants were classified as caregivers if they reported that they had provided practical assistance to someone with a long-term illness, disability or frailty for, at least, 3 hours a week in the past 12 months. Binary variable. | 1.Depressive symptoms Measured using the Center for Epidemiological Studies Depression (CES-D) index. 2.Anxiety Measured using the Geriatric Anxiety Inventory short form (GAI-SF) | Providing informal care was associated with increased depression (but not anxiety) compared to not providing care. [Depression: B = 0.49 95% CI (0.04, 0.95) $p<0.05$, Anxiety B = -0.03 (-0.20, 0.14)] | 8 stars |
| Amerio 2021 | Northern Italy | Undergraduate students aged 18+ 8177 participants (1191 CG, 6986 NC) | Cross-sectional Data from a large web-based cross-sectional survey in 2020 conducted in a university institute in Milan, Northern Italy. | *Home caregiving* Participants were classified as caregivers if they nominated that they were caring for a person at home during the confinement. Binary variable. | 1.Depressive symptoms Measured using the Personal Health Questionnaire (PHQ-9) 2. Anxiety Measured using the Generalised Anxiety Disorder Scale (GAD-7) | Caring for a person at home was positively associated with mental health symptoms (both depression and anxiety), compared to not caring for a person at home. [Depression: OR = 1.75 95%CI (1.01 ,1.63) for males and 1.28 (1.32,2.31) for females. Anxiety: OR = 1.43 (1.14–1.79) for males and 1.49 (1.24, 1.79) for females.] | 4 stars |
| Beach 2021 | United States | Adults 18+ 3509 participants (576 CG, 2933 NC) | Cross-sectional Data from a 2020 online survey conducted with the University of Pittsburgh, University Center for Social & Urban Research (UCSUR) research registry panel members of Pittsburgh region residents (with a national supplement). | *Family caregiving* Participants were classified as family caregivers if they recorded a positive response to the question: "are you currently providing unpaid care to a spouse, parent, child, other relative, partner, or friend to help them take care of themselves because of a chronic illness or disability?" Binary variable. | 1.Depressive symptoms 2. Anxiety Both measured using the Patient-Reported Outcomes Measurement Information System (PROMIS)-29 Profile v2.0 four-item short forms for anxiety, depression, fatigue, sleep disturbance, and ability to participate in social roles and activities. | Family caregiving was associated with increased depression and anxiety compared to non-caregivers. [Depression B = 0.46, SE = 0.15 <0.001. Anxiety B = 0.76, SE = 0.16 $p<0.001$; from supplementary files] | 7 stars |
| Costi 2023 | UK | Persons 16+ 4698 participants (349 existing carers, 1655 new carers and 2694 never-carers) | Longitudinal Data from the UK Household Longitudinal Study (Understanding Society), including waves 8 (2016–18), 9 (2017–19 and 10 (2018–20) collected before COVID-19, together with eight COVID-19 surveys specifically designed to collect relevant information on a monthly or bi-monthly basis (from April 2020 until March 2021). | *Caregiving (outside of own home)* Participants were classified as caregivers if they recorded a positive response to the question: "do you provide some regular service or help for any sick, disabled or elderly person not living with you? New carers were defined as those who started caregiving after the COVID-19 outbreak, as recorded by the related question in Wave 1 of the COVID surveys (April 2020). Categorised as: 1) existing carers 2) new carers 3) never-carers. | Psychological distress Measured with the 12- item version of the General Health Questionnaire (GHQ12). | Caregiving was positively associated with psychological distress for new carers at 5 out of 8 time points over 2 years of pandemic (as per estimates below), whereas caregiving was generally not associated with psychological distress for existing carers, except for COVID wave-7 (B = 0.476, SE = 0.244, $p<0.1$) [GHQ-12 new carers; COVID waves: 1 (April 2020) (B = 0.369, SE = 0.137, $p<0.01$); 2 (May 2020) (B = 0.320, SE = 0.136, $p<0.05$); 3 (June 2020) (B = 0.386, SE = 0.138, $p<0.01$); 4 (July 2020) (B = 0.181, SE = 0.132, ns); 5 (Sept 2020) (B = 0.193, SE = 0.130, ns); 6 (Nov 2020) (B = 0.266, SE = 0.137, $p<0.1$); 7 (Jan 2021) (B = 0.246, SE = 0.140, $p<0.1$); 8 (March 2021) (B = 0.092, SE = 0.138, ns)] | 9 stars |
| DiGessa 2022 | UK | Grandparents 50+ 2468 participants (1188 no childcare, 1280 childcare pre-pandemic, then estimate is for the 256 who completely stopped caregiving during pandemic (NC) compared to CG reference of 453 with same or increased care) | Longitudinal Data from the English Longitudinal Study of Ageing, using pre-pandemic data from Wave 9 (2018/2019) and the second COVID-19 sub study (November/December 2020). | *Grandparental childcare provision* Participants were classified as grandparental caregivers if they had at least one grandchild aged 15 or younger and they looked after any of their grandchildren without the pandemic present before the pandemic. Those who did were then asked whether the amount of care provided to grandchildren changed during the pandemic compared to pre-pandemic levels. Categorised as: 1) No grandchild care pre-pandemic 2) Mostly same or increased 3) Mostly decreased or interrupted 4) Completely stopped. | Depressive symptoms Measured using the Center for Epidemiological Studies Depression (CES-D) index. | Compared to grandparents who mostly maintained unchanged their grandchild care provision, those who stopped altogether (non-carers) were more likely to report poorer mental health (OR = 2.04 95% CI 1.30, 3.19, $p<0.01$), even accounting for pre-pandemic health. | 10 stars |

*(Continued)*

**Table 1.** (Continued)

| First author (year) | Country | Analytical sample | Study design (for our estimate of interest) and data source | Exposure/s and measurement | Mental health outcomes | Main findings (All estimates reported are from adjusted models where available) | Quality assessment (NOS) |
|---|---|---|---|---|---|---|---|
| Fusar-Poli 2022 | Italy | Adults 18+ 1203 participants (565 CG, 638 NC) | Cross-sectional Data from an anonymous online questionnaire in April/May 2020 spread through social networks using a snowball technique. | *Family caregiving* Participants were classified as family caregivers if they acknowledged being the primary caregivers of a family member with a mental, relational, or physical disability living in the same household during the lockdown. Binary variable. | Psychological distress Measured with the 12-item version of the General Health Questionnaire (GHQ12). | Mean scores obtained at the GHQ-12 were significantly higher in caregivers than non-caregiving controls, indicating greater levels of psychological distress (Cohen's d = 0.71, p < 0.001) | 5 stars |
| Ganadjian 2022 | Latin America and the Caribbean | Adults 18 + 12,328 (1,242 caregivers, 10,873 non-carers with no disability, 213 people with disabilities) | Cross-sectional Data from a 2020 web-based survey conducted by PAHO: Alcohol Use during the COVID-19 pandemic in Latin America and the Caribbean (33 countries) | *Family caregiving* Participants were classified as family caregivers if, as part of a single self-reported disability question: "do you or any child or adult you live with have a physical, mental or intellectual/developmental disability?", they nominated yes to a child or adult. Categorised as: 1) Yes, I have a disability 2) Yes, a child or adult I live with (placing them into the family caregiver's category) 3) No to both (non-carer/no disability) | Anxiety Measured using the Generalised Anxiety Disorder Scale (GAD-7) | Compared to non-carers with no disability, there was no difference in anxiety level/symptoms for family caregivers. With no anxiety symptoms as reference, estimates were as follows: mild anxiety (OR = 0.93, 95% CI 0.79, 1.10, p = 0.456), moderate anxiety (OR = 1.12, 95% CI 0.90, 1.39, p = 0.286), severe anxiety (OR = 0.99, 95% CI 0.77, 1.27, p = 0.987). | 6 stars |
| Hammarberg 2020 | Australia | Adults 18+ 13,829 (1205 of which were caregivers of relatives, 2664 of which were caregivers or children) | Cross-sectional Data from a short, anonymous online survey of people living in Australia was launched 4 days after the COVID-19 restrictions were implemented in 2020. | *Caregiving for relatives* Doing unpaid work caring for dependent relatives *Caregiving for children* Doing unpaid work caring for children Both were binary variables. | 1. Depressive symptoms Measured using the Personal Health Questionnaire (PHQ-9). 2. Anxiety Measured using the Generalised Anxiety Disorder Scale (GAD-7) | *Caregiving for relatives* increased the risk of depression for all except men aged >50 years and increased the risk of anxiety for all but men aged <50 years. [Depression: for 19-49-year females OR = 1.52, (95%CI 1.21; 1.91), males OR = 1.8 (95%CI 1.02; 3.19); for ≥50 years females OR = 1.55 (95%CI 1.26; 1.91), males OR = 1.47 (95%CI 0.89; 2.44). Anxiety: for 19-49-year females OR = 1.34 (1.06; 1.69), males OR = 1.5 (0.8; 2.8); for ≥50 years females OR = 1.49 (1.19; 1.87), males OR = 2.32 (1.38; 3.9)] *Caregiving for children* increased the risk of depression for women aged >50 years and decreased the risk for younger women and increased the risk of anxiety for women aged >50 years but not for younger women. The effect on men of caring for children was not significant for depression or anxiety. [Depression: for 19-49-year females OR = 0.83 (0.72; 0.95), males OR = 1.21 (0.82; 1.78); for ≥50 years females OR = 1.33 (1.05; 1.67), males OR = 1.15 (0.66; 2.03). Anxiety: for 19-49-year females OR = 0.99 (0.86; 1.14), males OR = 1.13 (0.73; 1.75); for ≥50 years females OR = 1.34 (1.05; 1.73), males OR = 1.07 (0.57; 2.01] | 6 stars |
| Hung 2021 | Hong Kong | Females 18+ 417 participants (238 CG, 179 NC) | Cross-sectional Data from a 2020 online questionnaire, with participants recruited through the Hong Kong Federation of Women's Centre's, followed by purposive and snowball sampling. | *Primary home caregiving* Participants who were classified as caregivers had to be the primary caregivers of a family member with a mental, relational, or physical disability living in the same household during the lockdown. Binary variable. | 1. Depression 2. Anxiety Both were measured using the Chinese Depression Anxiety Stress Scales (DASS 21), comprising three constructs: depression, anxiety, and stress. | Compared to non-carers, being a primary caregiver was not significantly associated with a higher level of depression (B = 0.25, SE 0.91, ns), or anxiety (B = 0.92, SE 0.83, ns). | 4 stars |

*(Continued)*

**Table 1.** (Continued)

| First author (year) | Country | Analytical sample | Study design (for our estimate of interest) and data source | Exposure/s and measurement | Mental health outcomes | Main findings (All estimates reported are from adjusted models where available) | Quality assessment (NOS) |
|---|---|---|---|---|---|---|---|
| Landi 2022 | Italy | Young adults 18-29yrs 1823 participants (365 CG, 1458 NC) | Cross-sectional Data from an online survey in Jan/Feb 2021during second Italian mandatory lockdown, with young adult participants recruited from the general community through social media and snowball sampling. | *Family caregiving* Participants were classified as caregivers based on whether they were living with anyone with a serious physical or mental health condition. If 'yes', they indicated who was ill, with two variables subsequently created: parental illness and other ill family member. Those who reported no family member with a serious health condition were labelled the non-carer group. Categorised as; 1) parental illness, 2) other ill family member, 3) non carers. | 1.Depressive symptoms Measured using the Personal Health Questionnaire (PHQ-9). 2. Anxiety Measured using the Generalised Anxiety Disorder Scale (GAD-7) | Compared to non-carers, young adult carers reported greater depression and anxiety, in both the parental illness (PI) and other ill family member (OIFM) groups. [Depression: for PI β = 0.125, Cohen's f2 = 0.016, p<0.001; for OIFM β = 0.078, Cohen's f2 = 0.006, p<0.01. Anxiety: for PI β = 0.114, Cohen's f2 = 0.013, p<0.001; for OIFM β = 0.068, Cohen's f2 = 0.005, p<0.01] | 6 stars |
| Liu 2021 | China | Adults 18+ 2858 participants (1056 CG, 1802 NC) | Cross-sectional Data from a 2020 online survey called the psychological status of Chinese Adults during COVID-19. | *Family caregiving* Participants were classified based on their responses to binary questions regarding care of elders and care of children in the past 2 months. Categorised as; 1) care for the elderly only, 2) care for children only, 3) care for both the elderly and children, 4) no care. | Depressive symptoms Measured using the Center for Epidemiological Studies Depression (CES-D) index. | Caring for both the elderly and children was significantly associated with higher depressive symptoms compared with non-caregivers (B = 2.6, 95% CI: 1.3, 3.9, p<0.001). However, compared to non-caregivers, caring for the elderly only (B = 1.3, 95% CI: -0.1, 2.6), and caring for children only (B = 1.1, 95% CI: -0.1, 2.4) was not significantly associated with depression. | 7 stars |
| Mak 2022 | Canada | Adults 18+ No. participants varied depending on time point. 1st time point n = 10,306 participants (2535 CG, 7771 NC) | Longitudinal Data from the UK COVID-19 social study run by University College London, utilising data from 6 time points over the pandemic (commencing in March 2020). | *Informal caregiving* Participants were classified as caregivers if they reported that they had caring responsibilities for elderly relatives or friends, people with long-term conditions or disabilities, or grandchildren. Binary variable. | 1.Depressive symptoms Measured using the Personal Health Questionnaire (PHQ-9). 2. Anxiety Measured using the Generalised Anxiety Disorder Scale (GAD-7) | Informal caregivers experienced higher levels of depressive and anxiety symptoms than non-carers across much of the pandemic. [Depression: for 1st lockdown estimated average treatment effect (ATT) = 0.45, 95% CI (0.12, 0.78)**, easing of 1st lockdown ATT = 0.55 (0.27, 0.84)***, 2nd lockdown ATT = 0.78(0.18, 1.38)* 3rd lockdown ATT = 0.70(-0.06, 1.46) easing of 3rd lockdown ATT = -0.09 (-0.76, 0.58), end of restrictions ATT = 1.01(0.44, 1.59)**. Anxiety: for 1st lockdown estimated average treatment effect (ATT) = 0.27, (-0.03, 0.57), easing of 1st lockdown ATT = 0.42 (0.17, 0.67)**, 2nd lockdown ATT = 0.84 (0.33, 1.35)**, 3rd lockdown ATT = 0.77 (0.05, 1.49)*, easing of 3rd lockdown ATT = 0.10 (-0.50, 0.70), end of restrictions ATT = 0.62 (0.06, 1.17)*] *Note: no key for p-values associated with *'s was provided in paper.* | 7 stars |
| McGarrigle 2022 | UK | Older adults 60+ 3670 participants (568 CG, 3102 NC) | Longitudinal Data from the Irish Longitudinal Study on Ageing (TILDA), utilising COVID-19 sub-study-wave 6 (2020) and waves 3 (2014), 4 (2016), 5 (2018). | *Caregiving* Participants were classified as caregivers based on whether they cared for someone during the COVID- 19 pandemic (spouse, children, grandchild, other relative, friend or neighbour). Data from Waves 3–5 were used to characterize caring in the pre-pandemic period. Transitions in caring status were then captured and modelled. Categorised as; 1) no caring or stopped caring, 2) continued to care, 3) new carer. | Depressive symptoms Measured using the Center for Epidemiological Studies Depression (CES-D) index. | Compared to not caring, being a carer (continued to care or new carer) during the COVID-19 pandemic was not associated with increased depressive symptoms. [Depression: Continued to care B = 0.33, 95% CI (-0.12, 0.79); New carer B = 0.18. 95% CI (-0.13, 0.49)] | 8 stars |

*(Continued)*

**Table 1.** (Continued)

| First author (year) | Country | Analytical sample | Study design (for our estimate of interest) and data source | Exposure/s and measurement | Mental health outcomes | Main findings (All estimates reported are from adjusted models where available) | Quality assessment (NOS) |
|---|---|---|---|---|---|---|---|
| Noguchi 2021 | Japan | Older adults 65+ 922 participants (80 CG, 842 NC) | Cross-sectional Data from secondary panel data collected through mailed surveys to community-dwelling older adults living in Minokamo City, a semi-urban area in Japan. (1st survey in March 2020, 2nd survey in October 2020). | *Family caregiving* Participants were classified as family caregivers based on the question: "do you care for your family currently?" (They also assessed caregiving role, severity of care recipient's needs, and increased caregiver burden during the pandemic). Exposure measured at T2. Binary variable. | Depression Assessed using the Two-Question Screen consisting of the following questions: (1) During the past month, have you often been bothered by feeling down, depressed, or hopeless? and (2) During the past month, have you often been bothered by little interest or pleasure in doing things? Those who answered yes to either or both questions were defined as showing depressive symptoms. Measured at T1 and T2. | Compared to non-caregivers, family caregivers were associated with the incidence (OR = 3.17, 95% CI 1.55–6.51, p < 0.01) and persistence of depressive symptoms (OR = 2.39, 95% CI 1.30–4.38, p < 0.01). *Note: While this paper adjusts for mental health measured at T1, the effect estimates presented here pertain to the cross-sectional association between informal care and mental health at T2.* | 6 stars |
| Park 2021 | United States | Adults 18+ 4,784 participants (689 short-term CG, 662 long-term CG, 3433 NC) | Longitudinal Data from the Understanding America Study, a nationally representative internet panel of more than 8,500 adults (utilising caregiving survey from January 2020 and a COVID-19 study from April/May 2020) | *Caregiving (short and long-term)* Participants were classified as caregivers based on the question: "in the past 30 days, did you spend any time assisting a family member or close friend (e.g., parent, grandparent, wife, husband, adult child, other family member, neighbour or close friend) with their basic personal activities? By that we mean daily activities such as dressing, eating, bathing, paying bills, managing medication, food preparation, grocery shopping, doctor visits, emotional support, driving, and other types of personal assistance." Then, based on a follow up question regarding duration of care, individuals were considered short-term caregivers (caregiving role for a year or less) or long-term caregivers (caring for greater than 1 year). Categorised as: 1) non-caregiver, 2) short- term caregiver, and 3) long-term caregiver. | Psychological distress Measured using the Personal Health Questionnaire (PHQ-4) | Both short-term (OR = 1.27, SE = 0.119, p<0.05) and long-term caregivers (OR = 1.27, SE = 0.121, p<0.05) are more likely to report psychological distress than non-caregivers. | 8 stars |
| Rodrigues 2021 | Austria | Adults 20–64 2000 participants (278 CG and 1722 NC) | Cross-sectional Data from AKCOVID, a representative survey carried out in Austria in June/July 2020 in combination with comparable 2015 data from the European Social Survey. *Note—Authors employed a "quasi-longitudinal design" whereby respondents were asked questions in retrospect to February 2020 (pre-onset) and in reference to the time of the survey (post-onset) in June/July 2020.* | *Informal caregiving* Participants were classified as informal caregivers if they reported providing care or help to sick, disabled, or frail family members, friends, or neighbours (but excluding care related to a paid job). Caregiving was assessed in relation to February 2020 (pre-onset of pandemic) and the time of the AKCOVID survey (post-onset). Binary variable. | Depressive symptoms Measured using the Center for Epidemiological Studies Depression (CES-D) index | The psychological well-being gap between carers and non-carers increased after the onset of the pandemic, the gap in CES-D scores widening by 2.5 units (Caregiving x post-onset b = 2.5, SE = 1.1, p<0.05). | 6 stars |
| Russell 2022 | United States | Adults 18+ 801 participants (176 CG, 625 NC) | Cross-sectional Data from online longitudinal, anonymous study of coping and risk behaviour during COVID (participants recruited from Amazon's MTurk worker pool), with two data collection points 60 days apart in 2020. Estimate for depression was cross-sectional data collected at baseline. | *Primary caregiver for child* Participants were classified as caregivers based on whether they were a [5] 18 or not. Binary variable. | Depression The Depression Anxiety Stress Scale-21 (DASS-21) | Non-caregivers reported significantly higher rates of depression symptoms than primary caregivers of children (t = -2.06, p< .05, d = 0.16). | 3 stars |
| Whitley 2021 | UK | Adults 16+ 9735 participants (565 CG, 9170 NC) | Longitudinal Data from the UK Household Longitudinal Study (UKHLS -Understanding Society) 2019 wave and the April 2020 and July 2020 COVID-19 surveys. *Note–UKHLS data is also used by Costi 2023, however Whitley examines home caregiving whilst Costi examines caregiving outside the home (different CG samples).* | *Home caregiving* Participants were classified as home carers based on the question: "is there anyone living with you who is sick, disabled or elderly whom you look after or give special help to?" Binary variable. | Psychological distress Measured with the 12- item version of the General Health Questionnaire (GHQ12). | Compared to non-caregivers, home caregiving had increased odds of psychological distress at both time points (April 2020 OR = 1.51, 95% CI 1.20, 1.90 and July 2020 OR = 1.66, 95% CI 1.24, 2.22), and suffered a greater decline in mental health during the pandemic (Difference in mental health score April to July 2020 = 0.79, 95% CI 0.33, 1.25). *p values not reported* | 6 stars |

(*Continued*)

**Table 1.** (Continued)

| First author (year) | Country | Analytical sample | Study design (for our estimate of interest) and data source | Exposure/s and measurement | Mental health outcomes | Main findings (All estimates reported are from adjusted models where available) | Quality assessment (NOS) |
|---|---|---|---|---|---|---|---|
| Wilson 2022 | UK | Adults 18+ 302 participants (182 CG, 120 NC) | Cross-sectional Data from an online survey during the first national UK COVID-19 lockdown in 2020. Survey distributed through social media, online message boards, charities, the Join Dementia Research forum and Call for Participants. | *Informal caregiver* Participants were classified as caregivers if they were not in a paid caring role (except for any state benefits/financial support for carers), had been in a caring role for 6 months or more, for someone aged 18+ who has a long-term condition commonly associated with caregiving (i.e., dementia, cancer, multiple sclerosis, and any mental health condition). Binary variable. | 1.Depressive symptoms Measured using the Personal Health Questionnaire (PHQ-9) 2. Anxiety Measured using the Generalised Anxiety Disorder Scale (GAD-7) | Compared to non-caregivers, caregivers demonstrated higher levels of depression (t = 6.47, p<0.001, d = 0.74) and anxiety (t = 4.92, p<0.001, d = 0.57). | 5 stars |
| Yoshioka 2021 | Japan | Persons 15–79 25,482 participants (2500 CG, 22,982 NC) | Cross-sectional Data from the Japan COVID-19 and Society Internet Survey (JACSIS) conducted in Aug/Sept 2020. | *Family caregiving* Participants were classified as caregivers based on whether they were caregiving for an elderly/disabled family member or not. Binary variable. | Psychological distress Measured with the Kessler Screening Scale for Psychological Distress (K6) | Compared to non-caregiving, caregiving to an elderly/disabled family member was associated with serious psychological distress in both women (OR = 3.00, 95% CI 2.12, 4.25, p<0.001), and men (OR = 6.67, 95% CI 3.96, 11.23, p<0.001). | 6 stars |

rated as high RoB [48, 53, 55]. Domains most likely to dampen quality (limit star allocations) were those relating to non-response, comparability, and temporality. In general, the domains of representativeness of the exposed cohort and selection of the non-exposed cohort rated well across studies, and assessment of outcome ubiquitously (and inevitably) met star requirements for all studies due to our inclusion criteria specifying that studies must contain a validated measure of MH.

### 3.3. Data synthesis

To complement the narrative synthesis, Table 2 presents an effect direction plot [39] providing visual synthesis of the effect direction of all included studies in the review. Moreover, to aid interpretation and enable more nuanced data synthesis, three Albatross plots [40] (Fig 2A–2C) present different subgroup estimates based on exposures and outcomes of included studies. These were:

*a) Adult caregiving and depression*
Eleven studies [44, 45, 47–50, 52, 56–58, 60] (comprising 14 estimates) presented sufficient data to calculate a standardised mean difference (SMD) or Cohen's d, allowing comparison and construction of an albatross plot for the association between adult caregiving and depression (Fig 2A). All studies show a positive effect size; nine estimates falling between SMD contours of 0.15 and 0.05, three estimates between 0.25 and 0.15, and one below 0.05. This indicates a moderate positive association between caregiving and poorer mental health, with no definitive gender subgroup differences.

*b) Adult caregiving and anxiety*
Six studies [47–49, 51, 52, 60] (comprising 8 estimates, two of which were unadjusted) [47, 49] presented sufficient data to construct an albatross plot for adult caregiving and anxiety (Fig 2B). All studies showed a positive effect size; five estimates falling between SMD contours of 0.15 and 0.05, and three null results with SMD<0.05. This indicates a small positive association between adult caregiving and anxiety.

*c) Childcare and depression*
Whilst few studies in the review examined childcare [42, 52, 54, 55], we constructed an albatross plot (Fig 2C) for those with available estimates [42, 52, 55] to allow visual representation of the opposing effect direction reported for this small sub-group. Notably, whilst one contributing study to this negative result had the lowest risk of bias (10 stars) [42], another had the highest risk (3 stars) [55]. Therefore, whilst this plot indicates a small negative association between childcare and depression, it must be interpreted with caution given the minimal contributing data, and presumably biased data for one estimate.

### 3.4. Narrative synthesis

Of the twenty studies included in this systematic review, the overwhelming majority reported a negative association between care provision during the pandemic and MH. Of the nineteen studies that examined depression/psychological distress as a MH outcome, thirteen studies uniformly reported higher levels of depression/psychological distress amongst caregivers compared to non-carers [43–45, 47–50, 52, 56–60], whilst one reported a negative association only for new carers (and none for existing carers) [41], and another only for sandwich care (and none for eldercare or childcare only) [54]. Two studies reported no association [46, 53], whilst two studies reported a decrease in depressive symptoms associated with care provision [42,

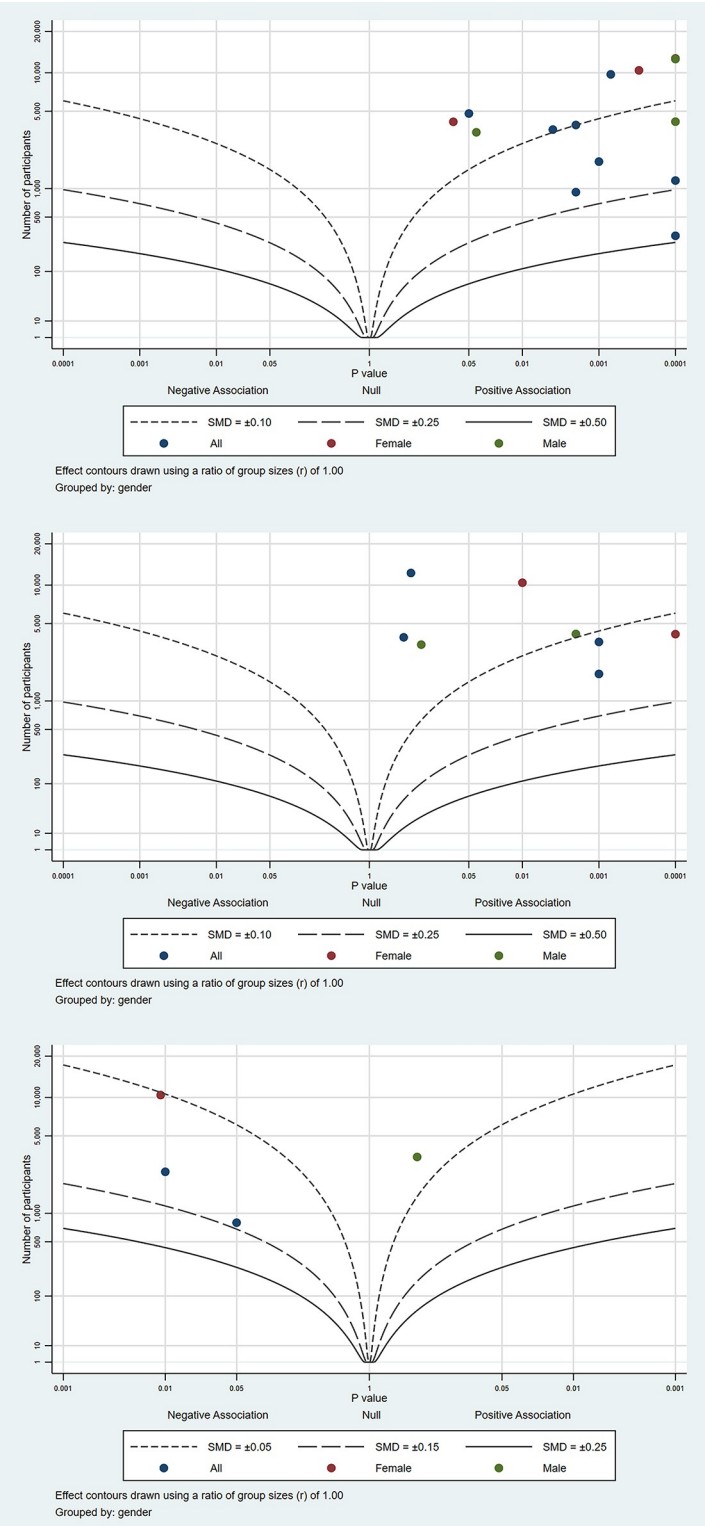

**Fig 2. a.** Albatross plot—Adult caregiving and depression. **b.** Albatross plot—Adult caregiving and anxiety. **c.** Albatross plot—Childcare and depression.

**Table 2. Effect direction plot/table.**

| First author, year | | Quality assessment (NOS) | Caregiving exposure (compared to no care) | Effect measure | Depression/ psychological distress | Anxiety |
|---|---|---|---|---|---|---|
| *Longitudinal* | | | | | | |
| DiGessa 2022 | Longitudinal | 10 stars | Categorical (no grandchild care pre-pandemic/ mostly same or increased/mostly decreased/ completely stopped). | OR | ▼ (Grandparental CG compared to those who stopped caring) | - |
| Costi 2023 | Longitudinal | 9 stars | Categorical (existing carers, new carers, never-carers). | B | ▲New carers ◄►existing carers | - |
| McGarrigle 2022 | Longitudinal | 8 stars | Categorical (no caring or stopped caring, continued to care, or new carer). | B | ◄► | - |
| Park 2021 | Longitudinal | 8 stars | **Categorical** (non-caregiver, short-term caregiver, or long-term caregiver). | OR | ▲ | - |
| Mak 2022 | Longitudinal | 7 stars | Binary | ATT | ▲ | ▲ |
| Whitley 2021 | Longitudinal | 6 stars | Binary | OR | ▲ | - |
| *Cross-sectional* | | | | | | |
| Allen 2022 | Cross-sectional | 8 stars | Binary | B | ▲ | ◄► |
| Beach 2021 | Cross-sectional | 7 stars | Binary | B | ▲ | ▲ |
| Liu 2021 | Cross-sectional | 7 stars | Categorical (care for elderly only, care for children only, care for both the elderly and children, no care). | B | ▲Caring for both elderly and children ◄► Caring for only elderly or only children | - |
| Ganadjian 2022 | Cross-sectional | 6 stars | Categorical (I have a disability/a child or adult I live with (has a disability (CG)/neither (NC/no disability). | OR | - | ◄► |
| Hammarberg 2020 | Cross-sectional | 6 stars | Binary | OR | ▲ | ▲ |
| Landi 2022 | Cross-sectional | 6 stars | Categorical (parental illness, other ill family member, non carer). | β | ▲ | ▲ |
| Noguchi 2021 | Cross-sectional | 6 stars | Binary | OR | ▲ | - |
| Rodrigues 2021 | Cross-sectional | 6 stars | Binary | B | ▲ | - |
| Yoshioka 2021 | Cross-sectional | 6 stars | Binary | OR | ▲ | - |
| Fusar-Poli 2022 | Cross-sectional | 5 stars | Binary | d | ▲ | - |
| Wilson 2022 | Cross-sectional | 5 stars | Binary | d | ▲ | - |
| Hung 2021 | Cross-sectional | 4 stars | Binary | B | ◄► | ◄► |
| Amerio 2021 | Cross-sectional | 4 stars | Binary | OR | ▲ | ▲ |
| Russell 2022 | Cross-sectional | 3 stars | Binary (carers of children) | d | ▼ | - |

| Key | | | |
|---|---|---|---|
| **Type of association reported** | | **Measure of effect used** | |
| upward arrow ▲ | Increased mental health impact (e.g., higher depressive or anxiety symptoms) | B/β | Coefficient of linear regression/standardised coefficient |
| downward arrow ▼ | Decreased mental health impact, (e.g., lower depressive or anxiety symptoms) | OR | Odds ratio |
| sideways arrow ◄► | No association/mixed effects/conflicting findings | d | Cohen's d |
| - | Not examined | Estimate | Effect estimate "type" not stated in the paper |

Lastly, to complete the data synthesis, Fisher's metanalyses of p values [38] was conducted for all included studies in the review, suggesting strong evidence of unpaid caregiving being associated with poorer MH overall (p<0.001).

55]. However, both these studies differed from other included studies in the review in that caregiving was for children, grandchildren in one study [42], and primary care for children in the other [55]. A third study reported a decrease in depression for childcare for women aged under 50 years, but an increase in depression and anxiety for women over 50 [52]. Eight studies examined anxiety as a MH outcome, with five reporting caregiving to be associated with higher levels of anxiety compared to non-caring [43, 48, 49, 50, 60], and three reporting no association [47, 51, 52].

## 4. Discussion

To our knowledge, this is the first systematic review to synthesise and assess the quality of the quantitative evidence examining the association between unpaid caregiving and MH during the pandemic. By imposing a requirement for a non-caring comparator, our review accounted for the effect of the pandemic to interrogate the relationship between informal caregiving and MH during this period. Our results indicate that, in the pandemic setting, unpaid caregiving was detrimental to MH, particularly depression and psychological distress. Fifteen out of the twenty included studies reported poorer MH in caregivers compared to non-carers (either overall, or in at least one category when caregiving was interrogated as a categorical variable). Three studies reported no association. An overall positive association was reported in two studies examining childcare. The overall quality of the evidence was moderate.

Prior to the COVID-19 pandemic, informal unpaid care provision was known to impact not only the lives but also the MH of those who provide it [17, 18]. Whilst the COVID-19 pandemic was unprecedented in the demand it placed on caregivers, paid and unpaid alike, it was unclear whether the MH of unpaid caregivers remained poorer compared to non-carers. This review has answered this question. In line with Carbone et al.'s review of dementia caregivers specifically [29], our review found that informal unpaid caregiving had a detrimental impact on mental health in the context of the pandemic. This is important, as whilst the WHO recently declared an end to COVID-19 as a global health emergency [61], response readiness for emerging variants and predicted future pandemics is vital [62]. Part of this preparedness should be to ensure that key learnings pertaining to the unique challenges faced by informal unpaid caregivers during the pandemic are prioritised, and aid in driving better policy in anticipation of future public health emergency events.

Our overall finding of poorer MH amongst caregivers during the pandemic is supported by other research aligned with this topic. This is predominantly via published work documenting caregivers' experiences during COVID-19, which illustrate feasible pathways to the reported increased prevalence of depressive and anxiety disorders found by our review. For example, a US based survey of caregivers during 2020 showed that most carers reported an increase in stress (83%) and a greater feeling of loneliness (77%) during the pandemic [63]. This aligns with Dellafiore et al.'s earlier rapid systematic review reporting caregivers' heightened stress as a key theme [28]. Moreover, a 2022 systematic review of the qualitative literature reported not only a rise in caregiving demands for carers, but a notable increase in caregiver negative emotions such as fear and uncertainty [64]. Importantly, factors such as knowledge of possible asymptomatic transmission of the virus and worries regarding no visitation rights if their loved care-recipient was hospitalised exacerbated these fears. Moreover, constant uncertainty was fueled by changeable and often abrupt and oscillating loss of health services as well as postponement/interruption of care recipient treatment, and, for some carers, economic insecurities was intensified. The same qualitative review also highlighted the decreased availability of social support, leading to concerns about carer wellbeing and ability to cope [64]. On top of

this, in another study, carers described having to provide even greater emotional support than usual to their care recipients, subsequently reporting heightened mental exhaustion [65].

Therefore, in considering these recurring themes of caregiver stress, fear, uncertainty, and loneliness, coupled with decreased support, we reflect that the pandemic delivered somewhat of a perfect recipe for caregiver's increased susceptibility to poorer MH. Whilst the mechanisms through which informal caregiving may impact MH in "normal times" are documented in detail elsewhere [15, 16, 18, 66], it is likely that the caregiving stress process model (SPM) [13, 67] also aptly describes many of the possible mechanisms at play during COVID-19. According to the SPM, it is the dual mix of objective circumstances and environment serving as primary stressors, as well as a carer's subjective (secondary) experience of such stressors that contribute to MH effects [13, 67], both of which were heavily influenced by the pandemic. Another likely mechanism (well-known prior to the pandemic) is the family effect, whereby the carer wellbeing is contingent on the health and welfare of the care recipient [68]. Research prior to the pandemic examining the family effect showed informal carers suffer adverse MH effects as the health of their parents, spouses or loved ones deteriorated [14, 69]. We hypothesise that this family effect is highly relevant, with carers witnessing the fear, distress, and loneliness of their care recipients as the threat of the pandemic and the consequences of its associated public health measures unfolded. Lastly, whilst the employment status of carers varied within and between the studies included in this review, we also postulate that, for those working, the juggling of work commitments with care provision remained challenging, with time poverty pressures and reduced time for self-care (exercise, recreation, relaxation, sleep) possibly also contributing to poorer MH [24, 70, 71]. Ultimately however, we posit that the pandemic not only intensified many of the challenges associated with caregiving in "normal times" but, amid throwing the entire global population into turmoil, had serious and deeply troubling *additional* effects on unpaid carers, the extent of which may take many years to fully understand.

In contrast to our overall findings, a small minority of studies included in this review reported a positive association between caregiving and MH during the pandemic. All three pertained to caregiving of children as their exposure (one being grandparental care [42], another for workers at Amazon who were primary carers of a child <18 [55], and one for childcare by women under 50years (the same study finding the opposite effect in older women and no effect in men)) [52]. It may be that caring for (presumably healthy) children during COVID provided much needed company when most households/persons were extremely isolated due to the pandemic. It is also possible that children provided distraction and a greater sense of purpose during this time. Nonetheless, given the small number of studies and the heterogeneity of the exposure across them, this review cannot draw robust conclusions about the overall effect of care for children during the pandemic on MH. Moreover, our findings do not align with the 2021 systematic review by Panda et al. who reported that caregivers of children and adolescents developed anxiety and depression while being in isolation with children, but they did not measure care per se [30]. Of note, another study in our review (reporting no association for childcare or eldercare as stand-alone exposures), did find sandwich care (both child and elder care) to negatively impact mental health [54]. This aligns with other research demonstrating that balancing caregiving responsibilities of older or disabled adults, whilst also having childcare responsibilities was identified as an increased risk factor (increased subjective burden and a lower carer-related quality of life scores) for caregivers during COVID-19 [65].

Overall, our review and the associated supporting research suggests that the pandemic was a particularly bleak time for informal unpaid caregivers. Our findings point not only to an ongoing need to alleviate the psychological risks of informal caregiving in a post-pandemic world, but also shine a light on the disproportionate MH penalty paid by caregivers because of

the pandemic It is essential that these learnings ignite a concerted effort to instate policy and procedures that can be implemented in the face of future public health emergencies. Importantly, whilst we know that the pandemic accelerated the use (and acceptance) of digitally enabled psychosocial interventions [72], only limited work has been published regarding interventions initiated to support the caregiver MH during COVID to date [73]. A 2021 systematic review synthesised the available evidence and found only three studies (from China and Italy) that examined psychological support interventions for informal caregivers during the pandemic [74]. The review reported that all interventions were delivered in digital format, and that there was considerable heterogeneity between the type of treatment, digital service, and care recipients across the three studies [74]. Interventions included a cognitive and sensory–motor intervention designed to lower caregiver distress, a caregiver skills educational program, and telehealth psychological support services, with the review surmising interventions were effective in helping reduce caregiver distress and burnout while improving self-efficacy and wellbeing [74]. A 2023 systematic review of caregivers of people with dementia supported these findings, reporting that internet-based psychoeducational programs (comprising educational, psychological, and behavioural training) can improve some aspects of caregivers' MH and emotional wellbeing [75]. Ultimately, whilst some possible policy actions and recommendations (with the aim of improving outcomes for informal caregivers) have been described in detail elsewhere [76, 77], there remains a dearth of evidence to inform suitable and targeted support for caregivers in the event of a future pandemic or similar widescale emergency [73], a gap that needs to be urgently addressed before the window of information and learnings from the recent pandemic recedes and closes.

## 4.1. Limitations & strengths

This review has some limitations. Firstly, misclassification bias is possible due to both the exposure and outcome measures of included studies being self-reported through survey-based questionnaires. Whilst the MH outcome measures are much less concerning (given all were validated measures), the caregiving exposure measures were particularly susceptible, especially in those studies that presumed caregiver status based on household structure, rather than self-report. Nonetheless, this misclassification is likely non-differential and thus is expected to only bias results towards the null (potentially under-estimating the true effect of caregiving). Secondly, there is potential for reverse causation (bidirectionality) in the relationship between informal caregiving and MH and, given most of the included studies were cross-sectional in design, our review is limited with respect to drawing any causal inference. Thirdly, we acknowledge both the family effect (which could bias results away from the null) and the healthy carer effect, akin to the healthy worker effect (which could bias towards the null), neither of which could be adequately accounted for in our review. Furthermore, our review did not explore caregiving intensity or load (most included studies had caregiving cohorts that were largely blended), thus likely oversimplifying the experience of informal caregiving. Consequently, we concede this may have inadvertently masked any potential protective effect that lower levels of care provision may have on MH. Additionally, given most eligible studies were from high-income countries, our findings may not be generalisable to low-middle-income nations. Importantly, noteworthy strengths of this review include imposing a non-caring comparator, reporting according to strict PRISMA guidelines (registered PROSPERO), conducting a thorough quality assessment of RoB utilising an amended NOS tool, and employment of alternative (Cochrane recommended) data synthesis methods when a meta-analysis was not feasible.

Lastly, this review leaves some unanswered questions. Firstly, a lack of gender-stratified results in most included studies meant that we were unable to assess the potentially modifying

effect of gender in the association between caregiving and MH during the pandemic. This remains a significant gap, especially given a gender lens is also lacking in most of the pre-COVID literature in this space [17, 18]. A proposed scoping review examining the gendered impact of pandemic containment measures on unpaid care and MH in Europe may go some way to filling this gap once published [78]. Secondly, we were unable to truly evaluate the association between childcare and MH during the pandemic given so few studies had a non-caring comparator (fulfilling our review eligibility criteria). Future reviews could consider wavering this requirement and solely investigate the impact of childcare (especially in working-age populations) on MH during the pandemic.

## 5. Conclusion

This systematic review synthesised the quantitative evidence of the association between unpaid caregiving and MH during the COVID-19 pandemic. The findings demonstrate that the MH of informal carers, already poorer pre-COVID compared to non-caregivers, was disproportionally impacted because of the pandemic and its associated public health containment measures. Given that few studies examining childcare employed a non-caring comparator, further work is warranted to synthesise and assess the association between childcare and MH specifically. The overall quality of all included studies was moderate. Our review is timely as informal unpaid caregivers are increasingly being recognised as the forgotten or hidden care workers of the pandemic. Accordingly, this review provides important evidence highlighting the vulnerability of this group and should motivate actionable learnings and drive policies to ensure unpaid caregivers are better supported now, in the medium term, and crucially if, and when, another global public health emergency emerges.

## Supporting information

**S1 Checklist. PRISMA checklist.**
(DOCX)

**S1 File. Search strategy.**
(DOCX)

**S2 File. Quality assessment.**
(DOCX)

**S3 File. Data synthesis.**
(DOCX)

**S1 Table. Full text exclusions.**
(DOCX)

## Author Contributions

**Conceptualization:** Jennifer Ervin, Ludmila Fleitas Alfonzo, Yamna Taouk, Humaira Maheen, Tania King.

**Data curation:** Jennifer Ervin, Ludmila Fleitas Alfonzo.

**Formal analysis:** Jennifer Ervin, Ludmila Fleitas Alfonzo, Yamna Taouk.

**Investigation:** Jennifer Ervin.

**Methodology:** Jennifer Ervin, Ludmila Fleitas Alfonzo, Yamna Taouk, Tania King.

**Project administration:** Jennifer Ervin.

**Resources:** Jennifer Ervin.

**Software:** Jennifer Ervin.

**Supervision:** Tania King.

**Validation:** Jennifer Ervin, Yamna Taouk, Tania King.

**Visualization:** Jennifer Ervin, Yamna Taouk.

**Writing – original draft:** Jennifer Ervin.

**Writing – review & editing:** Jennifer Ervin, Ludmila Fleitas Alfonzo, Yamna Taouk, Humaira Maheen, Tania King.

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
