## [Decision Letter · Decision Letter 0]

27 Dec 2023

Unpaid Caregiving and Mental Health during the COVID-19 Pandemic - a Systematic Review of the quantitative literature

PONE-D-23-32331

Dear Dr. Ervin,

We’re pleased to inform you that your manuscript has been judged scientifically suitable for publication and will be formally accepted for publication once it meets all outstanding technical requirements.

Kind regards,

Rosemary Frey

Academic Editor

PLOS ONE

“This research was supported by an Australian Research Council (ARC) Linkage Project (LP180100035), and a Melbourne School of Population and Global Health Targeted Research Support Grant. JE and LFA acknowledge support from University of Melbourne Research Training Scholarships. TK is supported by an ARC Discovery Early Career Award (TK-DE200100607). YT is supported by a Victorian Health and Medical Research Fellowship. Funders did not have any involvement in the design or conduct of any part of the study.”

Reviewers' comments:

Reviewer's Responses to Questions

**Comments to the Author**

1. Is the manuscript technically sound, and do the data support the conclusions?

Reviewer #1: Yes

Reviewer #2: Yes

2. Has the statistical analysis been performed appropriately and rigorously? 

Reviewer #1: N/A

Reviewer #2: Yes

3. Have the authors made all data underlying the findings in their manuscript fully available?

Reviewer #1: Yes

Reviewer #2: Yes

4. Is the manuscript presented in an intelligible fashion and written in standard English?

Reviewer #1: Yes

Reviewer #2: Yes

5. Review Comments to the Author

Reviewer #1: PLos One - Manuscript Number: PONE-D-23-32331

This systematic review addresses a critical gap in the literature by examining the impact of the COVID-19 pandemic on the mental health of unpaid and informal caregivers. A very current and relevant review.

The aim of this review is relevant, timely, and important in providing guidance on how policymakers and stakeholders implement programmes to support the mental health and well-being of informal caregivers not only during any events of public health emergencies such as the pandemic but at all times. The method applied for this literature review is suitable for the review that led to insightful results that have provided relevant answers with sufficient evidence from the literature. The results of this review have provided sufficient answers to the research questions.

The authors have presented their work in a very clear and comprehensive manner. For example, the authors presented the results of their findings in a way that answers some questions that readers might want to ask such as how quality assessment and data syntheses.

I find this work to be perfect in the sense that it covers a wider and more diverse population of those who are mainly round-the-clock engaged in informal unpaid care jobs worldwide. Without this population group, there could be a global care disaster as a result of a lack of caregivers. Informal caregivers' general health and well-being, most especially their mental well-being must be enhanced by all means possible. This work is timely, relevant, and key to ensure key stakeholders and policymakers are charged to provide the required support this population group needs.

This work is commendable and the following points below were noted:

P6 L 17 - 24

A very good and clear presentation on the variability of the definitions of caregiving across diverse informal caregivers. This has/would enable future work to capture different aspects of informal caregiving that exist. Thank you for this.

P6 L 26 - 30

Another very good insight into how MH was perceived by different informal caregiver groups. Well done.

Continue from P6 L 32

32 3.2. Quality assessment

P7 L7 - L 32

A very good and elaborate presentation of results of validated measures of MH including various mental health conditions. This can be used to enhance future research.

P8 L6 - L7

Here we see very interesting contrasting results; a decrease in depression for childcare for women under 50 while recording increased levels of both depression and anxiety in women over 50. It would be interesting to know whether the ages of the children in childcare were contributing factors to these contrasting results.

This paper is honestly giving that long-sought-after voice to millions of voiceless vulnerable and invisible unpaid caregivers across the world.

Thank you so much for this.

Reviewer #2: An excellent paper describing a body of work that is both important and well conducted. I read through it carefully twice trying to find anything I could suggest to improve it, but could find nothing. I am recommending publication.

6. PLOS authors have the option to publish the peer review history of their article (what does this mean?). If published, this will include your full peer review and any attached files.

Reviewer #1: **Yes: **Muhammad Aledeh

Reviewer #2: No

---

## [Editor Report · Acceptance letter]

27 Mar 2024

PONE-D-23-32331 

PLOS ONE

Dear Dr. Ervin, 

I'm pleased to inform you that your manuscript has been deemed suitable for publication in PLOS ONE. Congratulations! Your manuscript is now being handed over to our production team.

Kind regards, 

on behalf of

Dr. Rosemary Frey 

Academic Editor

PLOS ONE